# A Fast and Robust Lane Detection via Online Re-Parameterization and Hybrid Attention

**DOI:** 10.3390/s23198285

**Published:** 2023-10-07

**Authors:** Tao Xie, Mingfeng Yin, Xinyu Zhu, Jin Sun, Cheng Meng, Shaoyi Bei

**Affiliations:** School of Automible and Traffic Engineering, Jiangsu University of Technology, Changzhou 213001, China; xietao8301997@163.com (T.X.); zhuxinyu060503@163.com (X.Z.); sun1731997@163.com (J.S.); chenm140214@163.com (C.M.); bsy@jsut.edu.cn (S.B.)

**Keywords:** lane detection, re-parameterization, attention mechanism, row anchor

## Abstract

Lane detection is a vital component of intelligent driving systems, offering indispensable functionality to keep the vehicle within its designated lane, thereby reducing the risk of lane departure. However, the complexity of the traffic environment, coupled with the rapid movement of vehicles, creates many challenges for detection tasks. Current lane detection methods suffer from issues such as low feature extraction capability, poor real-time detection, and inadequate robustness. Addressing these issues, this paper proposes a lane detection algorithm that combines an online re-parameterization ResNet with a hybrid attention mechanism. Firstly, we replaced standard convolution with online re-parameterization convolution, simplifying the convolutional operations during the inference phase and subsequently reducing the detection time. In an effort to enhance the performance of the model, a hybrid attention module is incorporated to enhance the ability to focus on elongated targets. Finally, a row anchor lane detection method is introduced to analyze the existence and location of lane lines row by row in the image and output the predicted lane positions. The experimental outcomes illustrate that the model achieves F1 scores of 96.84% and 75.60% on the publicly available TuSimple and CULane lane datasets, respectively. Moreover, the inference speed reaches a notable 304 frames per second (FPS). The overall performance outperforms other detection models and fulfills the requirements of real-time responsiveness and robustness for lane detection tasks.

## 1. Introduction

In recent years, with the rapid development of intelligent transport systems (ITS), they play a key role in traffic safety [1]. Among the features of these systems, lane detection technology has received widespread attention as an important component of assisted driving. Lane lines clearly delineate driving zones for various types of vehicles. This contributes to reduced road congestion and aids in collision avoidance, thus ensuring road safety [2].

In practical driving, the complexity and diversity of traffic scenarios challenge lane detection. For example, capturing the full shape of lane lines is difficult under conditions of dazzle or insufficient lighting. The thin and elongated appearance of lane lines makes them susceptible to obscuration by surrounding vehicles. Drivers need timely feedback about road conditions. Thus, driver assistance systems must swiftly ascertain the location of lane markings. A formidable challenge in this area is to achieve a balance between lane detection accuracy and real-time responsiveness.

Lane detection technologies can be classified into two main categories: one relying on conventional image processing techniques, and the other on deep learning approaches. Traditional lane detection algorithms primarily use computer vision techniques alongside image processing methodologies to discern the color [3,4], texture [5,6], and other features of lane lines against the surrounding road surface. Algorithms like Sobel [7] and Canny [8] are employed to extract the boundaries of lane lines. Furthermore, by incorporating methodologies such as the Hough Transform [9,10] or Random Sample Consensus (RANSAC) [11,12] can serve to further augment the optimization of detection results. For instance, Cai et al. [13] suggested using a Gaussian Statistical Color Model (G-SCM) to extract areas of interest based on lane line color characteristics. This was then combined with an improved Hough Transform for lane detection within the extracted image region. Guo et al. [14] suggested combining an improved RANSAC version with the Least Squares method to optimzie model parameters, achieving enhanced lane fitting results. However, traditional lane detection methods require manual feature selection and extraction. In intricate driving scenarios, these methods often struggle to discern clear lane lines. This is especially true in circumstances with an absence of structured lane lines or variable lighting conditions.

Contrary to the conventional lane detection algorithms, deep learning techniques can automatically extract and learn features, continually updating model parameters through training on large-scale datasets [15]. This narrows the gap between predictive outcomes and actual results, addressing the challenges of lane feature extraction in complex scenarios. Nevertheless, deep learning demands a vast volume of training data and high computational performance. Therefore, the complexity of the model requires thorough consideration for its practical applications.

Currently, deep-learning-based methods for lane detection consist of three categories: those founded on segmentation [16,17,18,19,20,21], parameter regression [22,23,24], and anchor-based methods [25,26,27,28]. Segmentation-based detection methods can be further divided into semantic segmentation and instance segmentation. Pixels are classified by semantic segmentation in order to identify lanes and backgrounds as separate categories. On the other hand, instance segmentation not only identifies the category of each pixel, but also distinguishes between different instances of objects, making it useful for detecting multiple lane lines, especially when their count varies. However, segmentation tasks typically involve extensive computation, posing challenges to the real-time requirements of driver assistance systems. Parameter regression-based methods use neural network regression to predict parameters. These parameters are then used to construct a curve equation representing the lane lines. While these algorithms can identify lane lines with changing shapes, their predictions are significantly influenced by regression parameters, leading to poorer model generalization. Row-anchor-based methods use prior knowledge of lane line shapes and divide the image into location grids oriented in the row direction. A classifier then returns grids containing lanes. Although this method provides relatively quick inference speeds, its accuracy might not always be optimal.

Building on prior work in lane detection and considering the requirements for real-time and accurate performance, this paper refines the row-anchor-based detection methods. Initially, using the concept of online re-parameterization [29], we convert the multi-branch convolution into a single-structured online re-parameterization convolution. Using this convolution in the network not only ensured detection accuracy, but also reduced both training cost and inference time. Considering detection accuracy, we design a hybrid attention module that combines positional and channel attention mechanisms. This module can capture the spatial relationships and channel dependencies in image information, thereby enhancing the performance of the network. The main contributions of this paper are outlined as follows:We propose a lane detection model that integrates online re-parameterized ResNet and row-anchor classification. This model possesses efficient inference speed, ensuring real-time detection under various complex traffic scenarios.A hybrid attention module combining position and channel attention is designed, which captures feature information more comprehensively, enabling the model to focus on the slender lane line details in the image.Comparative experiments are performed on the TuSimple and CULane datasets with other lane detection models. Our model achieves better detection results. The experiments demonstrate that the proposed model meets the accuracy and robustness requirements for lane detection.

## 2. Related Work

### 2.1. Lane Detection Based on Deep Learning

To cope with the complex and ever-changing driving scenarios, researchers have applied deep-learning-based feature extraction methods to lane detection. Neven et al. [17] present the LaneNet model, which consists of an embedding vector branch and a semantic segmentation network. This model employs an encoding–decoding operation to transform input images into high-dimensional feature vectors and back to the original image, successively determining whether each pixel belongs to the lane line. Seeking enhanced semantic information extraction capabilities, Pan et al. [18] introduced an original network architecture, SCNN, which incorporates a spatial convolution layer to facilitate both vertical and horizontal information propagation. The convolution layer contains connections in four directions: left, right, up and down, thereby enhancing the correlation of long-distance spatial information. However, the overall structure of the model is complex, requiring substantial computational resources and time. Consequently, the training and inference processes are significantly time-consuming. Hou et al. [19] incorporated Self-Attention Distillation (SAD) into Convolutional Neural Networks (CNNs). This innovative method facilitates knowledge distillation between different layers, enabling efficient utilization of information from varying layers to capture critical feature information. It is important to note that while SAD is only involved in the training phase and does not increase inference time, it inevitably escalates the computational cost of model training. Tabelini et al. [22] designed a parameter-based lane line detection model, PolyLaneNet, which represents lane line shapes through polynomial curves. As a regression model, it boasts a faster detection speed compared to segmentation models, but its refining ability is inadequate, and the detection precision is lacking. Qin et al. [27] suggested a row-anchor-based lane detection method, transforming pixel-level classification into global row selection classification, thus reducing the computational load during the inference process. However, due to the simplicity of the network architecture, the lane detection results may be somewhat deficient. Tabelini et al. [25] proposed an anchor point-based lane detection method. This method extracts features from each anchor point using feature maps generated by the main network and then combines these features with the global ones produced by the attention module. As a result, the model can connect information from multiple lanes, improving its detection accuracy compared to other anchor-based lane line detection methods.

### 2.2. Re-Parameterization

With the continuous development of CNNs, a series of high-precision models have emerged. These models often have deeper layers and more complex modules to achieve better prediction and recognition capabilities. However, the complexity of these models frequently leads to significant computational resource consumption, making real-time inference challenging. To enable models to achieve faster inference speeds while maintaining high precision, a strategy based on structural re-parameterization has been widely adopted. For example, ACNet [30] utilizes asymmetric convolution to construct the network, improving the robustness of the model to rotational distortion without increasing the computational cost of deployment. The RepVGG [31] model features different structures in its training and inference phases. During training, the model leverages a multi-branch topology structure to capture information at multiple scales. In contrast, during inference, it employs a single-branch architecture reminiscent of VGG [32], consisting of 3 × 3 convolutions and ReLU, to ensure efficient inference. The Diverse Branch Block (DBB) [33], a structure paralleling the Inception model, incorporates a multi-branch design. This design permits the substitution of any K × K convolution within the model throughout the training phase, capturing multi-scale features and thereby enriching the image information extracted.

### 2.3. Attention Mechanisms

The attention mechanism dynamically changes the weight of each feature in the image, mimicking the selective perception of the human visual system. It focuses on the critical areas of the image and suppresses irrelevant information. SENet [34] is the first to introduce attention into the channel dimension. It establishes the dependency relationship between convolutional feature channels through squeeze and excitation operations, allowing the model to learn to allocate weights to different channels and improve the utilization efficiency of important features. ECANet [35] is an adaptive channel attention mechanism. It does not depend on the full connection operation and focuses only on the cross-channel interaction of neighboring channels, reducing computational cost and memory consumption. To augment feature extraction, researchers consider the dependency relationships of channels and space, and design a fusion of different attention mechanisms. For example, CBAM [36] concurrently incorporates information from the primary dimensions of channels and spatial contexts, thereby empowering the network to extract more comprehensive features and enabling the network to extract more comprehensive features. DANet [37] designs parallel structure position attention modules and channel attention modules, enabling local features to establish rich context dependencies and effectively improving the detection results.

## 3. Proposed Method

Figure 1 illustrates a lane detection model that integrates online re-parameterization convolution and a hybrid attention module, which is primarily composed of an encoding network and a decoding network. The encoding network uses an online re-parameterization ResNet as its backbone to extract image features and incorporates a hybrid attention mechanism to enhance its focus on elongated lane line information. The decoding network processes the deep features from the encoding network, converts them into a flattened structure, and then feeds them into the Multi Layer Perceptron (MLP) classifier. This network utilizes a row-anchor classification detection method, classifying the image based on anchor points in each row, and ultimately outputs the lane line positions via the existence and localization branches.

### 3.1. Online Re-Parameterization

In response to the latency issues in real-time lane detection, we adopt a convolutional structure re-parameterization method to reduce detection time. The core idea involves employing multi-branch model structures during the training process. When transitioning to actual deployment, this intricate structure is condensed into a singular architecture through equivalent transformations. While this approach reduces the inference time without compromising the model’s performance, the model is quite complex during the training phase and thus has considerable training costs. To mitigate this, we further incorporate an online re-parameterization strategy in our model design. This allows the multi-branch structure to be reparameterized in real-time during the training phase, ensuring efficient operation even with limited hardware resources. The structure transformation of online re-parameterization is illustrated in Figure 2.

The online re-parameterization process unfolds in two stages: linearization and block squeezing. During training, the normalization layer is nonlinear, complicating the merging of intermediate layers into a singular convolutional layer. In the initial stage, all nonlinear normalization layers are removed, and in their place, linear scaling layers are introduced. Functionally akin to normalization layers, these scaling layers foster diversity in the optimization across different branches. To cap off the process, a normalization layer is reintroduced post-module, which serves to hasten convergence and stabilize the model throughout its training.

After linearization, block squeezing is carried out, simplifying the complex multi-branch topology into a single convolutional layer. We will now detail the simplification process. The operation of a two-dimensional convolution kernel can be described as follows:(1)Y=W∗X
where X and Y, respectively, denote the input and output tensors, and W denotes the weight of the convolution.

For a multi-branch topology, we first simplify the serial structure. Multiple sequential convolutional layers are represented as follows:(2)$$Y=W_{N}(W_{N-1}*\cdot \cdot \cdot (W_{2}*(W_{1}*X)))$$

In the sequential convolutional architecture, the channel count remains consistent. By leveraging the associative property of convolution, we can combine multiple convolutional layers into a single layer by first convolving the kernels. The conversion process can be illustrated as follows:(3)$$\begin{array}{ll} Y & =(W_{N}(W_{N-1}*\cdot \cdot \cdot (W_{2}*W_{1}))*X\\ & =W_{e}*X \end{array}$$
where $W_{e}$ denotes the end to end mapping matrix for the entire sequential structure.

In the subsequent steps, we simplify the parallel structure. Leveraging the linear superposition property of the convolution operation, the linear combination of multiple convolutions is equivalent to performing the linear combination first and then the convolution. The process is specifically demonstrated as follows:(4)$$Y=\sum_{m=1}^{M}\left(W_{m}*X\right)=(\sum_{m=1}^{M}W_{m})*X$$
where $W_{m}$ denotes the weight of the *m*-th branch, and $\sum_{m=1}^{M}W_{m}$ denotes the total weight of the convolutional network.

Based on the two simplification principles mentioned above, the complex multi-branch topology can be compressed into a single linear block. After this transformation, the complex module only requires one convolution, converting operations on the intermediate feature maps into operations on the convolutional kernel. This results in obtaining end-to-end mapping weights, thereby reducing the training cost. Consequently, complex multi-branch structures can be used in model design to ensure accurate detection and achieve faster inference speeds. As illustrated in Figure 3, this paper builds upon the DBB [33] structure by integrating a depthwise separable convolution branch [38] and frequency filter branch [39], leading to the creation of an online re-parameterization convolution module. This enhanced structure showcases notable generalization capabilities. Furthermore, the online re-parameterization convolution module replaces the standard convolution in ResNet18, and the improved network is termed OREP_ResNet.

### 3.2. Hybrid Attention Module

To more comprehensively capture the fundamental feature information in complex scenarios, we contemplate integrating an attention mechanism into the lane detection model [40]. We designed a Hybrid Attention Module (HAM) comprising the Efficient Channel Attention Module (ECAM) and the Positional Attention Module (PAM). By integrating these two mechanisms, the model is better equipped to attain a multi-dimensional feature representation, leading to heightened accuracy and enhanced robustness.

As illustrated in Figure 4, ECAM first applies Global Average Pooling (GAP) to the input feature map, resulting in a 1 × 1 × C feature vector. This operation reduces the size of the input features and effectively simplifies the complex spatial characteristics. The subsequent step is the processing of channel feature maps. ECAM replaces the Fully Connected Layer (FC) commonly found in traditional channel attention modules with a one-dimensional convolution operation. The output of the one-dimensional convolution is passed through the sigmoid activation function to obtain the weights for each channel. Notably, the size of this one-dimensional convolution kernel can be dynamically adjusted. The convolution kernel size is denoted as ‘k’, which signifies the number of adjacent channel information involved in attention prediction. Typically, dynamic kernel sizes are selected in accordance with the principle that layers hosting a larger quantity of channels necessitate a larger span of channel interaction, thereby employing a larger convolution kernel. Conversely, layers with fewer channels utilize a smaller kernel. The specific kernel size is dictated by the kernel adaptive Formula (5). Lastly, the 1 × 1 × C feature map is element-wise multiplied with the original H × W × C input feature map, realizing adaptive channel weighting.
(5)$$k=\psi (C)={\left|\frac{\log_{2}(C)}{\gamma }+\frac{b}{\gamma }\right|}_{odd}$$
where ${\left|t\right|}_{odd}$ denotes the closest odd number to t, C denotes to the number of channels, and γ and b dictate the ratio between the channel number and the convolution kernel size. Conventionally, the values selected for γ and b are 2 and 1, respectively.

As illustrated in Figure 5, PAM first convolves the input feature map $X\in R^{C\times H\times W}$ to produce three feature maps: A, B, and C. Each of these feature maps is subsequently reshaped into a structure denoted as $R^{C\times N}$ (N=H×W), Following this, the transpose of A is matrix-multiplied with B, and the resulting product is fed into a softmax layer to obtain $S\in R^{N\times N}$. This represents the correlation between two positions on the feature map, as detailed below:(6)$$S_{ji}=\frac{\exp(A_{i}\cdot B_{j})}{\sum_{i=1}^{N}\exp(A_{i}\cdot B_{j})}$$

Next, *S* is matrix-multiplied with *C*, and the resulting output is added to the original feature map to obtain the final positional weights, as represented below:(7)$$Y_{j}=\lambda \sum_{i=1}^{N}\left(S_{ji}C_{i}\right)+X_{j}$$
where λ denotes the increased weight during the learning process, starting initially at 0. Thus, different weights can be obtained for each position, realizing adaptive position weighting.

The final step is the fusion of the two attention output feature maps. This fusion process integrates the positional data derived from the global contextual information, while assigning channel weights enriched with more target features, making the model more sensitive to detail information.

### 3.3. Row Anchor Classification

In the decoding phase, given that the lane lines contained in the input images largely display continuous characteristics in the column direction in the actual detection process, and the lane lines, being slender in shape, occupy only a small area in the row direction, we use a detection method based on row anchor classification [41] to predict the position of the lane lines. Furthermore, to ensure the real-time performance of the model, we eliminated the segmentation branch during the decoding stage, thereby streamlining the model. As shown in Figure 6, the H×W image undergoes a grid division process to obtain h×(w+1) cells containing position information. Compared to the H×W×(n+1) classification computation required by segmentation method, the h×(w+1)×n classification computation required by row anchor classification is significantly reduced. This significant reduction in computational complexity enables the model to more effectively satisfy the real-time demands of lane detection.

In the context of row anchor classification, the mathematical formula utilized for predicting the lane lines in each cell is as outlined below:(8)$$P_{i,j}=f^{ij}(X)\;\;i\in \left[1,n\right],j\in \left[1,h\right]$$
where $P_{i,j}$ denotes the probability that each cell within the image contains lane lines, n denotes the total count of lane lines present in the image, and h denotes the quantity of row anchors.

### 3.4. Loss Function

First, we obtain the predicted values at each position using a classifier and calculate the loss for the predicted values compared to the true labels. The corresponding formula is as follows:(9)$$L_{cls}=\sum_{i=1}^{n}\sum_{j=1}^{h}L_{CE}(P_{i,j},T_{i,j})$$
where $L_{CE}$ denotes the cross entropy loss, and $T_{i,j}$ denotes the true class label.

Additionally, we add the loss calculation of the lane line existence branch, thereby filtering out the coordinate points without lane lines. The remaining coordinates constitute the lane lines, and the corresponding formula is as follows:(10)$$L_{\mathrm{ext}}=\sum_{i=1}^{n}\sum_{j=1}^{h}L_{CE}(E_{i,j},TE_{i,j})$$
where $E_{i,j}$ denotes the predicted existence value, and $TE_{i,j}$ denotes the true existence label.

To sum up, the total loss associated with the model is illustrated as follows:(11)$$L_{total}=\alpha L_{cls}+\beta L_{ext}$$
where α,β each denote different loss weight coefficients, and in the experiments, these loss weight coefficients are set to 1.

## 4. Experiment

### 4.1. Datasets

In this study, we focus our experimental design and analysis on two publicly available lane line datasets: TuSimple and CULane. The TuSimple dataset, which is widely used in the field of autonomous driving, consists of 6408 images of highway driving, encompassing various traffic conditions and road structures. The images in this dataset have a resolution of 1280 × 720 pixels, the high pixel resolution facilitates the detection of lane lines at greater distances during the training process. The CULane dataset, a large-scale dataset for general lane line detection, consists of 133,235 images annotated for lane lines with a resolution of 1640 × 590 pixels, covering a diverse range of lighting conditions, road types, and complex traffic environments. The detailed information about these two datasets is presented in Table 1.

### 4.2. Experimental Environment

The experimental environment is set up using the PyTorch 1.11.0 deep learning framework on the Ubuntu 20.04 operating system, complemented by Cuda 11.3. The hardware configuration includes an NVIDIA GeForce RTX 2080Ti graphics card and an Intel(R) Xeon(R) Platinum 8255C CPU @ 2.50 GHz. For model optimization, we employ the Stochastic Gradient Descent (SGD) algorithm with a batch size of 16, a learning rate of 0.01, and a weight decay coefficient of 0.0001.

### 4.3. Evaluation Indicators

The official CULane dataset provides the F1 score as a performance evaluation metric for models. The calculation formula is presented in Equation (12), calculates the Intersection over Union (IoU) between the true pixel values and predicted values of the lane lines. Predictions with an IoU greater than 0.5 are considered as True Positives (TP), while those with an IoU less than 0.5 are considered as False Positives (FP). Undetected lane segments are viewed as False Negatives (FN). The F1 score is the harmonic mean of precision and recall. A higher F1 score indicates an excellent overall performance of the model, signifying a reduced probability of false and missed detections.
(12)$$\left\{\begin{array}{l} Recall=\frac{TP}{TP+FN}\\ Precision=\frac{TP}{TP+FP}\\ F1=\frac{2\times Precision\times Recall}{Precision+Recall} \end{array}\right.$$

The official TuSimple dataset provides Accuracy, FP and FN as the primary evaluation indicators. ACC reflects the ratio of samples that are correctly predicted to the total number of samples. FP refers to instances where the model incorrectly categorizes a negative sample as positive, whereas FN signifies cases where actual positives are misidentified as negatives. A decrease in these two metrics indicates an enhancement in the model’s accuracy. The calculation formulas are as follows:(13)$$Accuracy=\frac{\sum_{clip}C_{clip}}{\sum_{clip}S_{clip}}$$
(14)$$FP=\frac{F_{pred}}{N_{pred}}$$
(15)$$FN=\frac{M_{pred}}{N_{gt}}$$
where $C_{clip}$ represents the number of accurately predicted lane points and $S_{clip}$ represents the total number of actual lane points. $F_{pred}$ is the number of erroneously predicted lane points, while $N_{pred}$ is the total number of predicted lane points. $M_{pred}$ stands for the number of lane points that were not predicted, and $N_{gt}$ denotes the genuine count of existing lane points.

Furthermore, in our experiments, the F1 score was also employed for the TuSimple dataset to provide a comprehensive evaluation of the overall capabilities of the model. Concurrently, FPS represents the number of frames the model can process within a single second. A higher FPS value signifies a swifter inference capability of the model, serving as a measure of its real-time performance.

### 4.4. Module Comparison Experiment

To demonstrate the effectiveness of online re-parameterization in reducing training costs, the ResNet18 model was tested both before and after the use of online re-parameterization during the training phase. The experiment assessed the number of floating-point operations (FLOPs), parameter quantity (Params), and params size, as shown in Table 2. Typically, FLOPs are employed to gauge the number of floating-point operations necessary for a model to execute a forward propagation. The introduction of online re-parameterization considerably curtails the volume of convolution computations, leading to a drastic drop in the FLOPs value, thereby economizing the computational resources requisitioned by the model. Moreover, there is a discernible reduction in the number of model parameters. The params size descends to 88% of the initial model, which means a reduced demand for memory throughout the training process, thereby facilitating successful model operation within constrained hardware resources.

To demonstrate the efficacy of the proposed hybrid attention mechanism, a comparison between individual attention mechanisms (PAM [37], ECANet [35]) and hybrid attention mechanisms (CBAM [36], HAM) was performed on the TuSimple dataset. As evidenced by Table 3, the hybrid attention mechanism, by taking into account feature dependencies across various dimensions, places a stronger emphasis on crucial information. This leads to a noticeable improvement in detection performance compared to individual positional or channel attention. Additionally, the HAM effectively captures long-distance contextual relationships. In the lane detection task, it achieves the best results across several Indicators compared to other attention modules.

### 4.5. Ablation Experiment

To substantiate the advantages of the improved lane detection model, the experiment designed detection schemes with different module combinations and calculated their respective F1 scores, ACC, and FPS values. The results for each metric are presented in Table 4. First, we replaced the ordinary convolutions in the ResNet model with online re-parameterization convolutions. Because the simpler convolution structure is used in the inference phase, the FPS value in the experimental results was significantly improved. The re-parameterization multi-branch structure in the training phase more effectively obtained deep feature information, resulting in a certain increase in model detection accuracy. The HAM was tested for its ability to focus on important information in images. The experimental results showed that the addition of the HAM improved both F1 and accuracy compared to the baseline model. but it also slightly increased the computational complexity, Compared to the baseline model, the final F1 score and accuracy increased by 0.7% and 0.5%, respectively, after combining the improvements from both modules. Moreover, the inference speed reached 304 FPS, indicating that the improved model achieved the objective of fast inference and high detection accuracy.

### 4.6. Performance Comparison of Different Models

This section presents a comparison of the test results between our model and other lane detection models on the TuSimple dataset, as detailed in Table 5. The comparison encompasses models such as segmentation-based detection models (LaneNet [17], SCNN [18], SAD [19]), polynomial regression-based detection model (PolyLaneNet [22]), and ranchor-based detection model (LaneATT [25], UFLD [27]).

Among these models, the segmentation-based detection model, which classifies each pixel to achieve accuracy, results in a marked increase in computation, thereby posing significant drawbacks in terms of detection speed. The model proposed in this study, on the other hand, employs an online re-parameterization structure during the encoding phase to simplify the inference process, and incorporates a row anchor classification strategy during the decoding phase, thereby achieving a noticeably superior detection speed compared to that of segmentation-based lane line detection models. The final model can reach an inference speed of up to 304 FPS. Furthermore, the inclusion of a hybrid attention module within the feature extraction network enhances the ability to focus on detailed features, leading to an improvement in accuracy by 0.53% compared to the anchor-based LaneATT model, and 2.74% compared to the polynomial regression-based PolyLaneNet model. Compared to the similar row anchor classification model UFLD, the model proposed in this study demonstrates improvements across all evaluation indicators. The experimental findings indicate that our model outperforms other advanced lane detection models in terms of F1 score, FN, and FPS. Overall, the improved model demonstrates substantial advantages in terms of the accuracy and real-time performance of lane line detection, ensuring strong competitiveness.

In order to provide a more intuitive demonstration of the performance of the proposed lane detection method, we have selected the segmentation model LaneNet and the row anchor classification model UFLD for a comparative visualization of the results.

As illustrated in Figure 7, all models perform well in the straightforward road scenes. However, in curved road detection, the LaneNet model experiences some drift, and the UFLD model fails to capture the curve trend. In contrast, our proposed model successfully detects the upcoming turns. When faced with near-field occlusion scenarios where lane markings are obscured by nearby vehicles, the LaneNet model exhibits a failure to identify the obscured lane markings, whereas the UFLD model only partially detects them. Our proposed model is able to completely identify the lane markings. Additionally, in situations with shadow occlusions, the other two models display less distinct detection markings, while our model maintains a desirable level of detection performance. In summary, when comparing results across various scenarios, our proposed model demonstrates a clear advantage over the other methods, characterized by high accuracy and low rates of missed detection.

### 4.7. Robustness Testing

During actual driving, the lane detection model faces the challenge of dealing with lane detection in different complex scenarios, which requires the model to have sufficient robustness. In this section, we conducted robustness testing experiments on the CULane dataset, comparing the detection results of our model with other lane detection models in nine different driving scenarios, including a comprehensive F1 score for detection in complex environments such as night, arrow, dazzle, and FP in the cross.

Table 6 lists the F1 scores for each lane detection method in different scenarios. The results show that our model achieved the best total F1 score. This represents an improvement of 13.8 over the LaneNet model and 0.9 over the UFLD model of the same type of lane anchor detection. The proposed model outperforms other lane anchor detection models in seven driving scenarios, including normal, congested and night conditions. However, as the model does not specifically address the challenges associated with strong light interference and the absence of lane lines at junctions, the overall F1 scores for these two scenarios are slightly lower than some algorithms. Despite some shortcomings, the overall experimental results validate the robust advantage of the improved model in detecting lane lines in various complex environments. The model can handle the majority of driving scenarios, thus meeting the robustness requirements for practical lane line detection.

To more intuitively reflect the detection performance of the model in different complex scenarios, Figure 8 visualizes the results of lane line detection across nine different driving scenarios from the CULane dataset. Within each category, the first row displays the original images from various driving scenarios, the second row indicates the lane line positions with red lines according to the actual labels, and the third row uses green lines to indicate the lane line positions as predicted by the model. As illustrated in the figures, compared to actual labels, the proposed model is capable of accurately detecting lane markings even in challenging conditions where the lanes are obscured by factors such as crowded and shadow. In scenarios with extreme variations in lighting conditions, such as dazzle or night, the model effectively discerns the lane positions, thereby enhancing driving safety. Furthermore, the model demonstrates ideal robustness by distinguishing other road surface features and extracting lane information even in other special scenarios.

## 5. Conclusions

To meet the real-time and robustness requirements of lane detection tasks, we present an advanced lane detection model that integrates online reparameterization of ResNet with a hybrid attention mechanism. By reparametrizing the ResNet structure, our model streamlines multi-branch topologies into a single branch structure. In comparison with other complex network architectures, this method not only reduces training overhead, but also boosts inference speed significantly, achieving an impressive 304 FPS, which surpasses current advanced lane detection algorithms. With the further inclusion of the hybrid attention module, our model forms an effective connection between spatial locations and feature channels, thereby improving the extraction of critical information. When tested on two public lane detection datasets, our model achieved F1 scores of 96.84% and 75.60%, showing outstanding detection performance across various challenging scenarios. In summary, our proposed method is ideally suited for real-time lane detection in complex environments. In future work, considering the model’s suboptimal detection results under dazzle conditions, we plan to investigate the addition of image preprocessing before the feature network to correct overexposed areas, aiming for better detection results in dazzle situations.

## Figures and Tables

**Figure 1 sensors-23-08285-f001:**
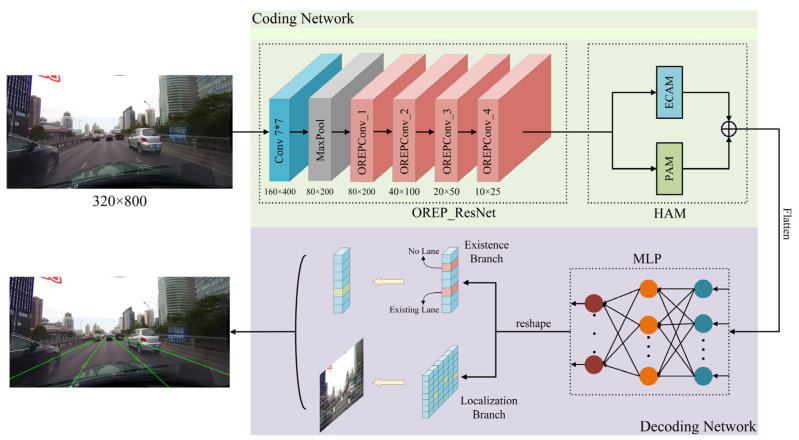
Overall structure of the lane detection model.

**Figure 2 sensors-23-08285-f002:**
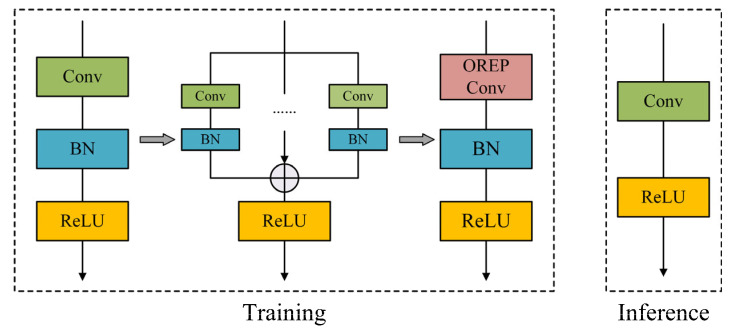
Online re-parameterization conversion process.

**Figure 3 sensors-23-08285-f003:**
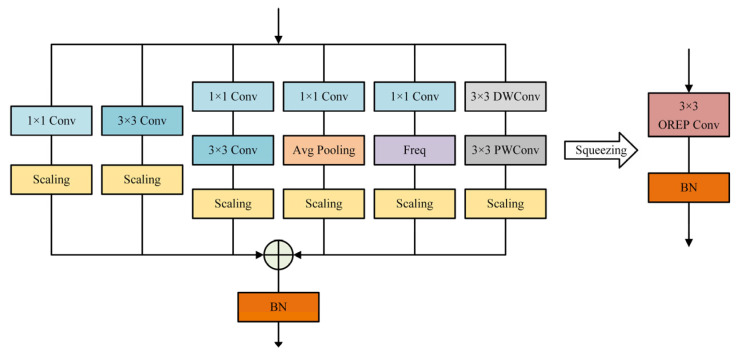
Online re-parameterization convolution module.

**Figure 4 sensors-23-08285-f004:**
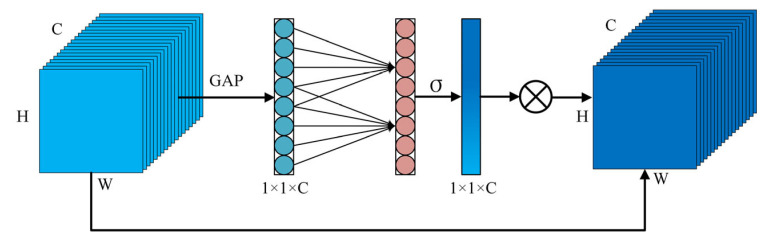
Detailed structure of the Efficient Channel Attention Module.

**Figure 5 sensors-23-08285-f005:**
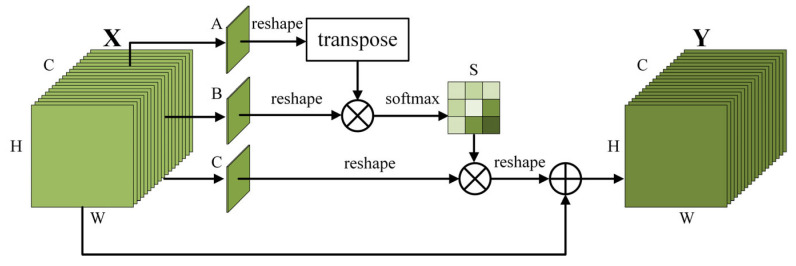
Detailed structure of the Position Attention Module.

**Figure 6 sensors-23-08285-f006:**
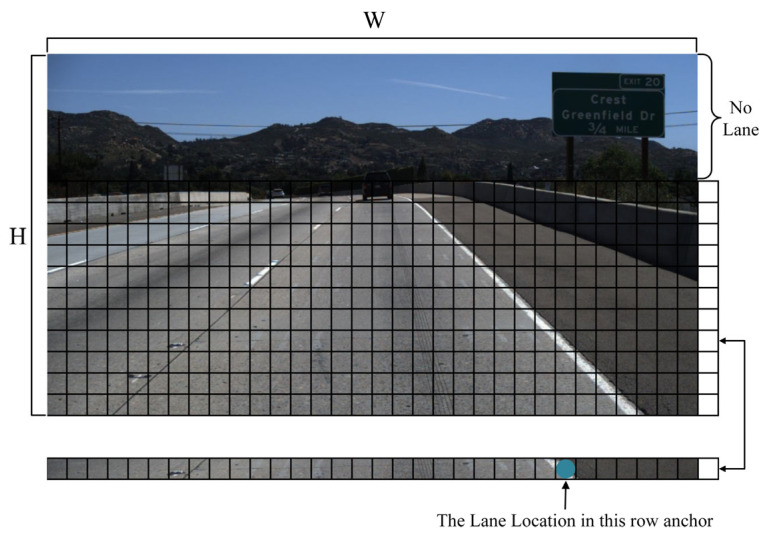
Row anchor classification diagram.

**Figure 7 sensors-23-08285-f007:**
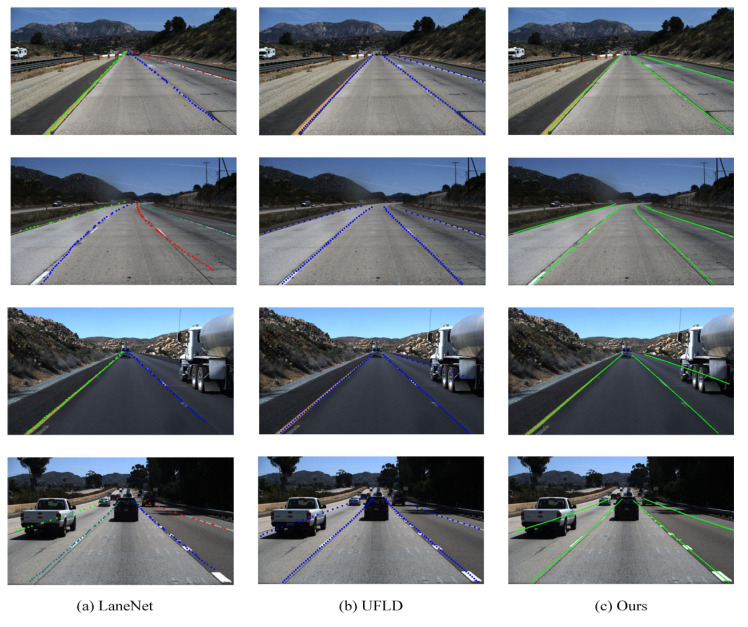
The detection results for the three models are presented. The first row is straight road scenes, the second row is distant curved road scenes, the third row is near-field occlusion scenes, and the fourth row is multiple occlusion scenes.

**Figure 8 sensors-23-08285-f008:**
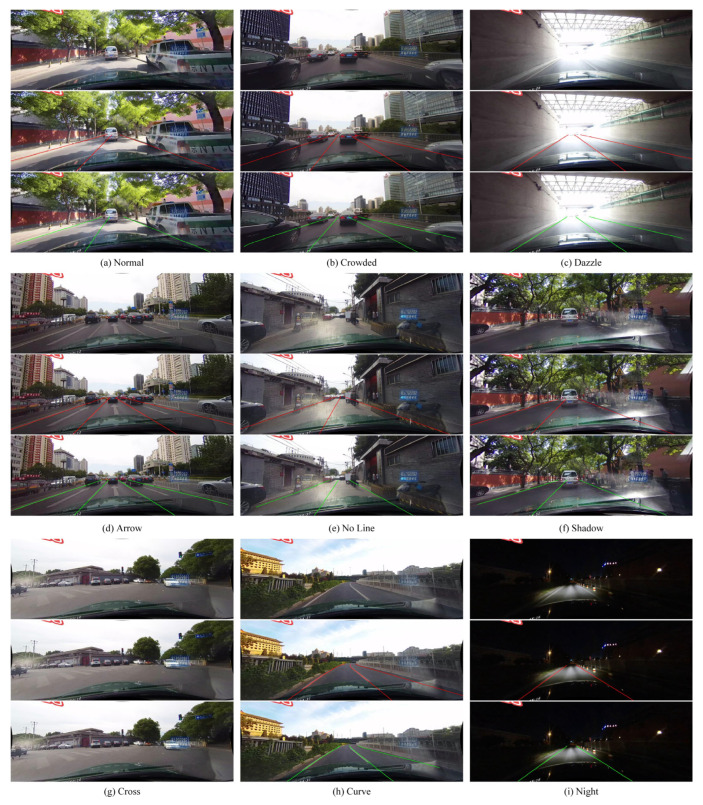
Lane detection visualization results across nine distinct traffic scenarios.

**Table 1 sensors-23-08285-t001:** Dataset information and partitioning.

Dataset	Frame	Train	Validation	Test	Resolution
TuSimple	6408	3268	358	2782	1280 × 720
CULane	133,235	88,880	9675	34,680	1640 × 590

**Table 2 sensors-23-08285-t002:** Results of computational volume evaluation metrics before and after the use of online re-parameterization.

Model	FLOPs/G	Params/M	Params Size/MB
Resnet18	9.389	96.369	367.62
Resnet_OREPA	0.235	85.375	325.68

**Table 3 sensors-23-08285-t003:** Attention module performance comparison. Bold numbers are the best.

Module	ACC	FP	FN	F1
PAM [37]	95.81	2.77	4.57	96.31
ECANet [35]	95.87	2.81	4.61	96.27
CBAM [36]	95.91	2.71	4.55	96.35
HAM	**96.03**	**2.68**	**4.28**	**96.50**

**Table 4 sensors-23-08285-t004:** Ablation experiment results. Bold numbers are the best.

Resnet18	OREP	HAM	F1	ACC	FPS
√			96.16	95.65	282
√	√		96.11	95.86	**338**
√		√	96.50	96.03	250
√	√	√	**96.84**	**96.10**	304

**Table 5 sensors-23-08285-t005:** Comparison of performance with different methods on TuSimple. Red, green, and blue numbers are the three results in descending order of optimality.

Method	F1	Acc	FP	FN	FPS
LaneNet [17]	94.80	96.38	7.80	2.44	44
SCNN [18]	95.97	96.53	6.17	1.80	7.5
SAD [19]	95.92	96.64	6.02	2.05	75
LaneATT [25]	96.71	95.57	3.56	3.01	250
PolyLaneNet [22]	90.62	93.36	9.42	9.33	115
UFLD [27]	96.16	95.65	3.06	4.61	282
Ours	96.84	96.10	2.29	4.00	304

**Table 6 sensors-23-08285-t006:** Comparison of performance with different methods on CULane. Red, green, and blue numbers are the three results in descending order of optimality.

Method	Normal	Crowded	Night	Noline	Shadow	Arrow	Dazzle	Curve	Cross	Total
LaneNet [17]	82.9	61.1	53.4	37.7	56.2	72.2	54.5	59.3	5928	61.8
SCNN [18]	90.6	69.7	66.1	43.4	66.9	84.1	58.5	64.4	1990	71.6
SAD [19]	90.1	68.8	66.0	41.6	65.9	84.0	60.2	65.7	1998	70.8
PINet [21]	85.8	67.1	61.7	44.8	63.1	79.6	59.4	63.3	1534	69.4
CurveLane [20]	88.3	68.6	66.2	47.9	68.0	82.5	63.2	66.0	2817	71.4
LaneATT [25]	91.1	72.9	68.9	48.3	70.9	85.4	65.7	63.3	1170	75.1
UFLD [27]	91.7	73.0	70.2	47.2	74.7	87.6	64.6	68.7	1998	74.7
Ours	92.1	74.1	71.3	48.4	77.1	88.3	63.1	69.3	1909	75.6

## Data Availability

Not applicable.

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
