# Peer review of "A Fast and Robust Lane Detection via Online Re-Parameterization and Hybrid Attention"

_sensors, 2023, doi:10.3390/s23198285_

Round 1

Reviewer 1 Report

The paper deals with a critical task for the perception of geographic information in driverless driving and advanced assistance. There are many methods known for lane line detection, but they have disadvantages because they require manual adjustment of parameters and have poor robustness.

The authors focused on a detection method based on residual network (ResNet) and row anchor classification.

For this reason, in the reviewer's opinion, Chapter 3.3 should be expanded to include a comparison of the proposed method with such methods described in the literature, for example:

A Novel Lane Line Detection Algorithm for Driverless Geographic Information Perception Using Mixed-Attention Mechanism ResNet and Row Anchor Classification. ISPRS Int. J. Geo-Inf. 2023, 12, 132.

Author Response

Dear reviewer:

Thank you very much for reviewing our manuscript amidst your busy schedule. Following your suggestions, we have made corresponding modifications to the manuscript.

Comment:

For this reason, in the reviewer's opinion, Chapter 3.3 should be expanded to include a comparison of the proposed method with such methods described in the literature, for example:

A Novel Lane Line Detection Algorithm for Driverless Geographic Information Perception Using Mixed-Attention Mechanism ResNet and Row Anchor Classification.ISPRS Int.J.Geo-Inf.2023,12,132.

Response:

Thank you for the suggestions you provided. As you mentioned, there was a lack of literature citation in section 3.3. The reference you provided is in line with our content, and we have added this citation to our manuscript. Moreover, compared to the method described in the reference, our approach removes the segmentation branch in the decoding part, making the model more concise.

We have made the following changes in section 3.3 on page 8 of the manuscript:

“we use a detection method based on row anchor classification[37] to predict the position of the lane lines. Furthermore, to ensure the real-time performance of the model, we eliminated the segmentation branch during the decoding stage, thereby streamlining the model.”

(Highlight in red in the manuscript)

Reviewer 2 Report

A very interesting and well presented research. While there is still a few minor language issue, the scientific mechanism is sound. For application and different and complex scenarios, a visualization for applying this model into complex scenario is essential for the argument.

However, the visualization of figure 7 is hard to read and need more elaboration for analysis. Otherwise, there is no much differences between second row and third row. 

I have no problem for reading this article

Author Response

Dear reviewer:

Thank you very much for reviewing our manuscript amidst your busy schedule. Your suggestions have been very helpful in improving the manuscript, and we have made the necessary modifications accordingly.

Comment:

However, the visualization of figure 7 is hard to read and need more elaboration for analysis. Otherwise, there is no much differences between second row and third row.

Response:

Thank you for your valuable suggestions. We indeed lacked a detailed analysis of the visualization in Figure 7 (The revised version is Figure 8). We contemplated visualizing the detection results in nine different scenarios and comparing them with the actual positions of the lane lines. We marked the actual labels in the second row for a better comparison, yet we did not provide further explanations of the content depicted in the images. Therefore, we have added the respective analysis on page 14 of the manuscript as follows:

“As illustrated in the figures, compared to actual labels, the proposed model is capable of accurately detecting lane markings even in challenging conditions where the lanes are obscured by factors such as crowded and shadow. In scenarios with extreme variations in lighting conditions, such as dazzle or night, the model effectively discerns the lane positions, thereby enhancing driving safety. Furthermore, the model demonstrates ideal robustness by distinguishing other road surface features and extracting lane information even in other special scenarios.”

(Highlight in red in the manuscript)

Reviewer 3 Report

The topic is an interesting one, but considering the technological advance exhibited by the field of communications and vehicle automation, as well as the infrastructure they benefit from, this approach does not bring anything new to the studied field.

The number of experiments or data that the authors present is not enough to convince the usefulness and necessity of the proposed solution. There are not enough arguments/comparisons between the implemented solution and other approaches.

What are the new elements brought by the proposed solution?

What are the applicability of the solution in relation to the subject studied?

Comparisons between what has been achieved in relation to other approaches in the specialized literature?

Online re-parameterization? Are the structures and layers for each software architecture made by you?

Can you show us the algorithm? The language? The schedule?

Can you provide reviewers with a GitHub folder or more extensive/complex results?

What are the most advantages that your approach brings in relation to what you have achieved?

Insufficient results and experiments are not clear.

The number of data you present in the table is not concrete, try several scenarios that will convince .

The conclusions and future directions are not sufficient and do not add value to the literature.

The references need improvement, updating, and more elements to clearly show that they are important for the literature.

No comments!

Round 2

Reviewer 1 Report

The changes presented, in the reviewer's opinion, may be satisfactory.

Reviewer 3 Report

The authors made an effort and it can be seen by the comments and adjustments made to the article.

Many of the observations made have benefited from extremely relevant answers, and in this way, I want to congratulate the authors.

The recommendation is that many of the answers given to the observations should also be found in the final material because it would be a shame if they were not exposed, and readers or other enthusiasts of the field would not benefit from these clearly exposed results.

The article has been properly adjusted, and there are no other observations, maybe only the images/captures should be a little more legible, bigger and clearer because in printed format the article is harder to follow, a certain color cannot be distinguished. detection, trajectory.

There are no other observations.